# Microneedle Delivery of Heterologous Microparticulate COVID-19 Vaccine Induces Cross Strain Specific Antibody Levels in Mice

**DOI:** 10.3390/vaccines13040380

**Published:** 2025-04-01

**Authors:** Tanisha Manoj Arte, Smital Rajan Patil, Emmanuel Adediran, Revanth Singh, Priyal Bagwe, Mahek Anil Gulani, Dedeepya Pasupuleti, Amarae Ferguson, Susu M. Zughaier, Martin J. D’Souza

**Affiliations:** 1Vaccine Nanotechnology Laboratory, Center for Drug Delivery Research, College of Pharmacy, Mercer University, Atlanta, GA 30341, USA; tanisha.manoj.arte@live.mercer.edu (T.M.A.); smitalrajan.patil@live.mercer.edu (S.R.P.); emmanuel.adediran@live.mercer.edu (E.A.); revanth.singh.sateesh@live.mercer.edu (R.S.); priyal.bagwe@live.mercer.edu (P.B.); mahekanil.gulani@live.mercer.edu (M.A.G.); dedeepya.pasupuleti@live.mercer.edu (D.P.); amarae.ferguson@live.mercer.edu (A.F.); 2College of Medicine, Qatar University, Doha P. O. Box 2713, Qatar; szughaier@qu.edu.qa

**Keywords:** SARS-COV-2, heterologous vaccine, microparticles, microneedles, humoral antibody response, Omicron, Delta variant, cross-reactivity

## Abstract

Background: In recent years, the COVID-19 pandemic has significantly impacted global health, largely driven by the emergence of various genetic mutations within the SARS-CoV-2 virus. Although the pandemic phase has passed, the full extent of the virus’s evolutionary trajectory remains uncertain, highlighting the need for continued research in vaccine development to establish a cross-reactive approach that can effectively address different variants. This proof-of-concept study aimed to assess the effectiveness of microparticulate vaccine delivery through the minimally invasive microneedle route of administration, using a heterologous prime–booster strategy against the SARS-CoV-2 virus. Method: This strategy uses the whole inactivated virus of the Delta variant for the prime dose and the whole inactivated virus of the Omicron variant for the booster dose, with alum as an adjuvant. The formulation of microparticles involves encapsulating the antigens in poly lactic-co-glycolic acid (PLGA) polymer, which provides sustained release and enhances immunogenicity while protecting the antigen. Microparticles were tested for in vitro assays, and characterization included particle size, zeta potential, and encapsulation efficacy. Furthermore, serum was collected post-administration of the vaccine in mice and was tested for antibody levels. Result: In vitro assays confirmed the non-cytotoxicity and the ability of microparticles to activate the immune response of the vaccine particles. Administering this microparticulate vaccine via microneedles has proven effective for delivering vaccines through the skin. We also observed significantly higher antigen-specific antibody levels and cross-reactivity in the strains. Conclusions: Our adjuvanted microparticulate-based heterologous prime–booster vaccine strategy showed cross-reactivity among the strains and was successfully delivered using microneedles.

## 1. Introduction

The COVID-19 pandemic, caused by the Severe Acute Respiratory Syndrome Coronavirus-2 (SARS-CoV-2) virus, led to around 776 million cases and 7.1 million deaths worldwide [1]. The virus originated in Wuhan, China, at the end of 2019 [2,3]. However, it was not until March 2020 that the World Health Organization (WHO) officially declared COVID-19 a pandemic, due to the increasing death rate and the growing number of positive cases globally.

Conventional antiviral treatments, including neuraminidase inhibitors, RNA synthesis inhibitors, and nucleoside analogs, were less effective against the virus. Despite the administration of antiviral drugs, convalescent plasma therapies, and monoclonal antibody treatments, vaccination remained the most effective strategy to curb the virus’s spread. The first vaccine, developed by Pfizer and Moderna in December 2020, was mRNA-based [4].

However, vaccines showed limited effectiveness against emerging variants of SARS-CoV-2, which resulted from significant mutations in the virus’s spike glycoprotein receptor-binding domain. Variants such as Delta, Omicron, Alpha, and Beta, which originated in different parts of the world, caused major disruptions during the second and third waves of the pandemic. The Delta variant, which originated in India around June 2021, became the most prominent strain, leading to more infections than any previous variant. It was approximately 60% more transmissible than the Alpha variant [5] (Figure 1)

The mutations in the S-protein domain of the spike protein enhanced viral transmissibility and reduced the effectiveness of convalescent plasma (CP) and monoclonal antibodies (MAbs). The Delta variant, characterized by mutations such as D614G, L452R, P681R, and T478K in the spike protein’s receptor-binding motif, led to increased replication of the virus in the upper respiratory tract and stronger interactions with ACE2 receptors. These changes reduced the neutralizing antibody response, making it a variant of concern worldwide [6,7]. A new variant arose, causing a rise in cases during the third wave. This variant, Omicron, underwent 50 mutations, including 32 unique spike protein mutations, giving it the advantage of stabilizing the binding moiety of the S-glycoprotein to interact with the ACE2 receptor, thereby amplifying the virus’s spread. Moreover, sub-variants like Omicron BA.4 and BA.5 are the most transmissible, causing more than 50% of total cases, with a rapid speed of invading the immune system and a higher viral load in the upper respiratory tract [8,9].

Various vaccination strategies were implemented, which used the whole inactivated viruses or different parts of viruses, such as the spike protein, nucleoprotein, mRNA vaccines, and vector-based vaccines. However, there is a need to develop a vaccine that provides broader immunity. Heterologous vaccine administration is one way to achieve this [10].

Heterologous prime–booster involves loading two different variants of the virus separately during the prime and booster phases of the vaccine regimen. This strategy is useful against mutating viruses, as a broader immune response is generated when the immune system is exposed to different variants. It has the potential to generate a cross-reactive response against other variants. Moreover, effective mass immunization is achievable in a short time span. Although this novel strategy for SARS-CoV-2 was previously used in influenza studies [11,12], in this proof-of-concept study, we have used whole heat-inactivated viruses of Delta and Omicron origin as our model antigens. Moreover, we plan to examine the immune response and cross-reactive response to the Alpha and Beta variants. The Alpha and Beta variants are early variants of the COVID-19 pandemic, where Alpha exhibited increased transmissibility and Beta showed immune evasion compared to the original strain. Testing cross-reactivity will determine the overall efficacy of the vaccine.

Conventionally, vaccines are delivered using intramuscular delivery; however, the need for trained physicians, lack of patient compliance, and potential risks of hematoma and muscle damage have led to the need for an alternative delivery strategy [13]. As research progresses in developing non-invasive vaccination routes, intranasal and intradermal delivery systems are being explored. Vaccines delivered intranasally have a major drawback regarding effective immune activation due to the limited ability of the vaccine to cross the mucosal barrier [14,15]. On the other hand, microneedle administration delivered through the skin is devoid of this limitation and might help eliminate vaccine hesitancy among individuals with needle phobia because of the use of hypodermic needles.

In our previous studies, the microparticulate vaccine delivered via microneedles has been shown to generate an excellent immune response for other vaccines, such as influenza, gonorrhea, Respiratory Syncytial Virus, Zika, and coronavirus [16,17,18,19,20]. Research has used poly (lactic-glycolic acid) (PLGA) as a polymer to formulate microparticles. The studies confirm that when polymers like PLGA are used as a matrix, they provide stability and prevent antigen degradation due to temperature. Moreover, the shape and size of PLGA-based microparticles govern the sustained delivery profile of the antigens. Microparticles with a size between 100 nm and 1000 nm are efficiently taken up by antigen-presenting cells (APCs) and produce a strong immune response [21]. Furthermore, as a biodegradable polymer, these vaccine formulations are non-toxic [22]. Thus, the PLGA polymer was used to encapsulate the SARS-CoV-2 antigen in this study.

We formulated microparticles using the double emulsion method. This technique involves an oil-in-water-in-oil phase, with the antigen in the aqueous phase and the polymer in the organic phase. The lyophilization technique was then used to remove excess water from the emulsion, yielding a dry powder. The dry powder form of microparticles has thermostability at elevated temperatures, thus eliminating the need for cold chain storage [23]. The adjuvant used in the vaccine microparticulate preparation will enhance the immune response by promoting co-stimulatory markers and antigen presentation, in turn activating naïve T-cells [24]. Alum was used as an adjuvant, which causes a depot effect that promotes immune cells and inflammatory monocytes at the site of administration. Moreover, it promotes a Th-2 biased immune response [25,26].

## 2. Materials and Methods

### 2.1. Materials

SARS-Related Coronavirus 2, Isolate hCoV-19/USA/MD-HP05285/2021 (Lineage B.1.617.2; Delta Variant), Gamma-Irradiated, and SARS-Related Coronavirus 2, Isolate hCoV-19/USA/GA-EHC-2811C/2021 (Lineage B.1.1.529; Omicron Variant), Gamma-Irradiated, were obtained through BEI Resources, NIAID, NIH. Poly(lactic-co-glycolic) acid (PLGA) 75:25 (Resomer^®^ RG 752H) was acquired from Evonik Industries ( Birmingham, AL, USA). Dichloromethane (DCM) was obtained from Fischer Scientific. Span^®^ 80 and trehalose dihydrate were purchased from Millipore Sigma (Burlington, MA, USA). 3-(4,5-Dimethylthiazol-2-yl)-2,5-diphenyltetrazolium bromide (MTT) and the Pierce Micro BCA Assay Kit were obtained from Thermo Fischer (Waltham, MA, USA). Murine dendritic cells (DC 2.4) were offered as a kind gift by Dr. Kenneth L. Rock (Dana-Farber Cancer Institute Inc., Boston, MA, USA). Adjuvants such as Alhydrogel^®^ (Alum) were purchased from Invitrogen (San Diego, CA, USA). A spring applicator was used for microneedle delivery. Horseradish peroxidase (HRP)-conjugated goat anti-mouse secondary IgA, IgM, IgG, IgG1, and IgG2a antibodies were acquired from Invitrogen (Rockford, IL, USA). 3,3′,5,5′-tetramethylbenzidine (TMB) was purchased from Becton, Dickinson & Co. (Franklin Laks, NJ, USA). Six- to eight-week-old Swiss Webster mice were obtained from Charles River Laboratories (Wilmington, MA, USA). Cell culture supplies, such as Dulbecco’s Modified Eagle’s Medium (DMEM), trypsin EDTA solution, fetal bovine serum (FBS), and penicillin/streptomycin, were acquired from the American Type Culture Collection (Manassas, VA, USA).

### 2.2. Formulating Microparticulate Microneedle Vaccine

Microparticles (MPs) were formulated using the (w/o/w) double emulsion method that was previously designed in our lab [27]. Firstly, the primary emulsion was formed, followed by the secondary emulsion. To formulate the primary emulsion, an aqueous phase comprising 2% loading of antigen, which is composed of the whole inactivated virus of Delta/Omicron, is mixed into an organic phase consisting of 2% *w*/*v* PLGA (polymer) in dichloromethane (DCM), along with span 80 as the surfactant. The mixture is then homogenized at 17,000 RPM for 2 min to form the primary emulsion. The secondary emulsion is formed by adding the primary emulsion to the aqueous phase, which consists of polyvinyl alcohol in distilled water. The mixture is homogenized and sonicated for size reduction. The secondary emulsion that is formed is placed on a magnetic stirrer for 5–6 h to allow for the evaporation of the DCM layer, followed by ultra-centrifugation for 15 min at 17,000 rpm at 4 °C. The microparticulate slurry settles at the bottom, which is then resuspended with a cryoprotectant consisting of 2% *w*/*v* trehalose and placed in a Labconco™ benchtop freeze dryer for lyophilization to obtain free-flowing dry powder. Microparticles of alum were formulated separately using the same process mentioned above, with 2% loading of alum into the polymer.

Microneedles were prepared by incorporating the vaccine microparticles and adjuvant microparticles in 10% sodium hyaluronate (HA) and 5% trehalose. Microparticles equivalent to each dose were calculated and weighed accordingly. A mixture of the previously weighed microparticles, along with HA, trehalose, and deionized water, was prepared in an Eppendorf tube to form a gel. This gel was then poured into an 8 × 8 array PDMS mold, with approximately 25 mg of evenly spread microneedle arrays across the surface. Using the spin casting method, the mixture was centrifuged at 3000 RPM for 15 min at 15 °C to ensure complete filling of the mold’s void spaces. The molds were left in a desiccator overnight to dry. The following day, a backing layer of pure HA was applied. After drying, the microneedles were carefully detached from the mold using double-adhesive tape (Figure 2). Moreover, a previous study from our lab shows a penetration depth of 520 μm in porcine skin. The depth of the penetration was measured using confocal microscopy using hematoxylin and eosin to visualize needle penetrability through skin layers. Dye used to measure pore-forming nature was analyzed using 1% methylene blue. Microneedles dissolve within 5 min after coming into contact with the skin [28].

### 2.3. Characterization of Vaccine Microparticles

#### 2.3.1. Percent Recovery Yield

Before and after the process of lyophilization, the weight of the microparticulate solution was noted. The following formula is used to calculate the practical yield:$$Percent\;recovery=\frac{Weight\;of\;MPs\;before\;Lyophilization-Weight\;of\;MPs\;after\;lyophilization}{Weight\;of\;solid\;ingredients\;from\;the\;formulation}\times 100$$

#### 2.3.2. Scanning Electron Microscopy

Approximately 0.5 mg of microparticles were dispersed in 1 mL of DI water along with 10 μL of Span 80. Around 10 μL of the mixture was spread on the stub and kept in a desiccator overnight. The next day, images were obtained, showing the morphology of the microparticles using a Phenom benchtop SEM, Nanoscience Instruments, Phoenix, AZ, USA.

#### 2.3.3. Particle Size, Count, Zeta Potential, and Poly-Dispersibility Index (PDI)

Zeta potential and PDI determine the charge and uniformity of the particles needed for effective delivery. A total of 1 mg of vaccine microparticles was resuspended in 1 mL of deionized water and transferred into the cuvette. A Zetasizer Nano ZS (Malvern Pananalytical, Westborough, MA, USA) was used to obtain size, zeta potential, and PDI, and a laser particle counter was used to determine the number of particles suspended per mL of medium.

#### 2.3.4. Encapsulation Efficacy

This assay uses a Bicinchoninic acid assay kit (BCA) to determine the antigen protein content in the microparticles. To achieve this, the polymer layer was dissolved by adding 2 mg of vaccine microparticles to dichloromethane (DCM). The mixture was centrifuged at 3000 RPM for 10 min at 4 °C. DCM was allowed to evaporate, and the residue obtained was dissolved in phosphate buffer and analyzed for protein content using a BCA assay. Concentration per mL was obtained by plotting the standard curve. The following formula is used to determine encapsulation efficacy:$$Encapsulation\;efficacy=\frac{Practical\;protein\;content\;in\;2\;mg\;of\;MPs}{Theoretical\;protein\;content\;in\;2\;mg\;MPs}\times 100$$

#### 2.3.5. Assessment of Antigen Integrity

The antigen integrity post-microparticle formulation was assessed using sodium dodecyl sulfate-polyacrylamide gel electrophoresis (SDS PAGE). Firstly, protein was extracted from the microparticles by adding dichloromethane to 5 mg of microparticles. This mixture was centrifuged at 17,000 RPM for 10 min at 15 °C and placed in the fume hood for the DCM layer to evaporate completely. Next, 500 μL of phosphate buffer was added, and the sample was loaded into the wells of a 7.5% Mini-PROTEAN^®^ TGX™ precast gel along with the sample buffer in a 3:1 ratio. Protein markers from Bio-Rad and the antigen suspension were used as the standards for the sample. The gel was placed in the running buffer at 120 V for 2 h. Using 0.1% Coomassie R-250 (Bio-Rad, Hercules, CA, USA), gel staining was carried out overnight, and it was de-stained with 50% methanol and preserved between cellophane sheets.

### 2.4. In Vitro Assessment of Microparticles

#### 2.4.1. Cytotoxicity Assessment Using MTT Assay

The 3-(4,5-dimethylthiazol-2-yl)-2,5-diphenyltetrazolium bromide assay (MTT) was used to determine the cytotoxicity of the formulation to murine dendritic cells (passage 2.4) in triplicates. Cells were grown in T-75 cell culture flasks, and around 10,000 cells were seeded into 96-well plates in triplicates and incubated for 24 h at 37 °C. The following day, cells were exposed to a microparticulate vaccine at different concentrations, such as (negative control-only cells) 5 μg/mL, 12.5 μg/mL, 25 μg/mL, 50 μg/mL, 100 μg/mL, 250 μg/mL, and 500 μg/mL, serially diluted from the stock and suspended in DMEM media. The stock was prepared by adding 2 mg of MPs to 1 mL of DMEM media. Positive controls included treating cells with a cytotoxicity reagent, dimethyl sulfoxide (DMSO). The plate was then incubated overnight at 37 °C. Finally, from each well, the media was removed, and 10 μL of 5 mg/mL MTT reagent, along with 90 μL of DMEM was added, which developed purple crystals after incubating at 37 °C for 2–3 h. Later, DMSO was added to dissolve the crystals, and the plate was read at 570 nm using a BioTek Synergy H1 plate reader (BIO-TEK Instruments, Winooski, VT, USA).

#### 2.4.2. Griess’s Assay

Griess’s assay was performed on murine dendritic cells (DCs) by exposing them to vaccine microparticles. DCs were grown to 90% confluency. Cells were detached by adding trypsin to the flask. Cells were plated in 96-well plates with a seeding density of 10,000 cells per well. The next day, cells were exposed to vaccine microparticles. For this purpose, 2 mg of MPs were suspended in 1 mL of DMEM, and the cells were exposed to a mixture consisting of 2 μg of whole inactivated vaccine microparticles of Delta, 2 μg of whole inactivated vaccine microparticles of Omicron, and 3 μg of alum. After incubating overnight, the supernatant was separated, 50 μL/well of supernatant was transferred to another 96-well plate, and 50 μL of 0.1% (1-naphthyl) ethylenediamine dihydrochloride (NED) reagent and 1% sulfanilamide were added to each well. A purple color appeared after incubation, and the plate was read at 540 nm using a BioTek Synergy H1 plate reader (BIO-TEK Instruments, Winooski, VT, USA). The concentration of the test sample was obtained after comparison with a nitrite standard reference curve. The nitrite standard curve was generated from the nitrite standards that were serially diluted, with dilutions of 0 μM, 3.13 μM, 6.25 μM, 12.5 μM, 25 μM, 50 μM, and 100 μM. The nitrite standard curve was obtained after plotting the average absorbance value with the nitrite standard concentration.

### 2.5. In Vivo Administration

Approval for all animal experiments was obtained from the Mercer University Institutional Animal Care and Use Committee (IACUC) protocol (#A2004006). The testing was carried out according to the approved protocol. Animal procedures were designed to minimize animal suffering and distress. This study adhered to the ethical principles of the 3Rs (Replacement, Reduction, and Refinement). Mice were housed in a controlled environment with free access to food and water and were monitored daily for signs of distress. To test vaccine efficacy, six Swiss Webster female mice, aged six to eight weeks, per study group were used in this study. Intramuscular administration was performed by administering the dose to the thigh muscle using a 26-gauge needle after restraining the animal. For microneedle administration, the dorsal side of the mice was used and shaved 24 h prior to administration so that hair interference did not affect dosing. The next day, mice were placed in an anesthesia chamber to restrict their movement and access the administration site. The skin was pulled over a small wooden block, and, using a microneedle spring applicator, effective administration was performed by holding the applicator over the skin. The groups included the naïve (no-treatment) group, the intramuscular group (antigen and adjuvant in suspension form), and the vaccine microparticulate microneedle (antigen microparticles and adjuvant) group. Mice were immunized with a prime dose consisting of a whole inactivated virus of Delta suspension, administered intra-muscularly and intra-dermally through MN administration on day 0/week 0, followed by a booster immunization of a whole inactivated virus of Omicron suspension at week 3, and were sacrificed at week 7 (Table 1). Blood and mucosal samples were collected bi-weekly after each dose. The tail-snip method was used to withdraw blood, and mucosal samples were collected from the vaginal area. Samples were spun down at 13,000 RPM for 10 min at 4 °C, and the supernatant was saved at −80 °C and later analyzed for antibodies using an enzyme-linked immunosorbent assay.

### 2.6. Detection of Antibody Response Using Enzyme-Linked Immunosorbent Assay

Enzyme-linked immunosorbent assay (ELISA) is an assay used to analyze antibodies present in mouse serum samples. In this study, mouse serum was tested for IgM, IgA, IgG, and subtypes like IgG1 and IgG2a against antigens such as whole-cell inactivated viruses of SARS-CoV-2 variants Delta and Omicron and spike glycoprotein RBD of Alpha and Beta variants. For the ELISA assays, high-binding 96-well plates were coated with coating antigen and incubated overnight at 4 °C. The next day, the plates were washed with 0.05% Tween in phosphate buffer saline (PBS) and blocked with 3% bovine serum albumin (BSA) in PBS for 3 h at 37 °C, followed by the addition of mouse serum samples previously diluted 1:100 in 1% BSA in PBS. The plates were incubated overnight at 4 °C. Depending on the antibody to be analyzed, goat anti-mouse secondary antibody (IgM, IgA, IgG, IgG1, IgG2a) conjugated with horseradish peroxidase (HRP) were added to the wells after washing the plates thrice. The plates were then incubated for 90 min at 37 °C, and tetramethylbenzidine (TMB) substrate was added. The plates were then incubated for 10 min to allow the color reaction to proceed. The reaction was stopped using 0.3 M H_2_SO_4_, and the plates were read by the BioTek Synergy H1 plate reader (BIO-TEK Instruments, Winooski, VT, USA) at 450 nm.

### 2.7. Statistical Analysis

Statistical analysis was performed using GraphPad Prism 10.3.1 software (GraphPad Software, San Diego, CA, USA). Data from physiochemical characterization, in vitro, and in vivo assays were expressed as mean ± SEM. A one-way ANOVA test, along with post hoc Tukey’s multiple comparison test, was conducted. Two-way ANOVA was used to test dependable groups. The following significance levels were used: ns, non-significant, * *p* ≤ 0.05, ** *p* ≤ 0.01, *** *p* ≤ 0.001, **** *p* ≤ 0.0001. A *p*-value of <0.05 is considered statistically significant.

## 3. Results

### 3.1. Characterization of Microparticulate Microneedles

The encapsulation efficacy of the microparticulate formulation was >85% for both microparticulate formulations containing the Omicron whole inactivated virus and the Delta whole inactivated virus. The average particle sizes for Delta MPs and Omicron MPs were 692.7 nm ± 38.7 and 731 ± 21.3 nm, respectively (Table 2).

### 3.2. SEM Imaging

Microparticle images observed under a scanning electron microscope (SEM) confirmed the spherical nature of the particles. Additionally, when the microneedles were scanned using the SEM, they demonstrated sharp pointed ends with an average length of 429.33 ± 11.5 μm (Figure 3).

### 3.3. In Vitro Assesssment of Microparticles

#### 3.3.1. In-Vitro Cytotoxicity Assessment Using MTT Assay

The MTT assay was performed to assess cell toxicity induced by the microparticulate vaccine. In this experiment, dendritic cells (DCs) treated with DMSO served as a positive control group to demonstrate toxicity. At all tested concentrations of the vaccine microparticles, cell viability remained above 90% compared to the untreated control group. Furthermore, the viability of both the untreated control group and the microparticulate vaccine-treated groups at all concentrations was significantly higher than that of the DMSO-treated positive control group. These results indicate that the microparticles are non-toxic to dendritic cells (Figure 4).

#### 3.3.2. Determination of Antigen Integrity

The integrity of the antigen was confirmed by analyzing and comparing the bands formed by Delta and Omicron microparticles with those formed by Delta and Omicron suspensions. After running SDS-PAGE and staining the bands with Coomassie Blue, visible and detectable bands were observed between 75 and 50 kDa in both the antigen suspension and the microparticles (Figure 5).

#### 3.3.3. Griess Assay

The Griess assay was used to quantitatively measure nitrite release from dendritic cells in response to different treatment groups. Omicron + Adjuvant MPs and Delta + Adjuvant MPs exhibited significantly higher nitrite release compared to their respective Omicron suspension, Delta suspension, blank microparticulate, and no-treatment group (Figure 6).

### 3.4. Analyzing Antibody Responses Using Enzyme-Linked Immunosorbent Assay (ELISA)

#### 3.4.1. Serum Antibody Response to In Vivo Administration

Mice serum samples were analyzed for IgM, IgG, IgA, IgG1, and IgG2a antibodies using indirect ELISA. When mice were vaccinated with both doses of the Delta and Omicron variants, along with an adjuvant intradermally, significant Delta-specific IgM antibodies and Omicron-specific IgM antibody responses were observed in the treatment group compared to the untreated group across week 2, week 5, and the terminal week. The cross-reactive response for Alpha-specific IgM antibodies was non-significant in the terminal week when the naïve group was compared to the treatment group. Beta-specific antibody levels in the microneedle treatment group were non-significant across all weeks, except in week 2. Additionally, when comparing the intramuscular group to the microneedle group, no significant differences were observed in Omicron-, Beta-, and Alpha-specific antibody responses throughout the study period. No significance was observed between the intramuscular and microneedle groups for the Delta-specific IgM antibody (Figure 7).

Antigen-specific IgA levels were significantly higher in the treatment group compared to the untreated group throughout the study period for the Delta, Omicron, Alpha, and Beta strains. Additionally, antibody levels peaked at week 2 for the Omicron strains and in the terminal week for the Delta strain. However, no significant differences were observed between the naïve and treatment groups for Delta- and Omicron-specific antibodies across the study period. Notably, the intramuscular suspension group exhibited a significantly higher response compared to the microneedle group at weeks 2 and 5 for the Alpha variant. In contrast, a significant difference was observed in the intramuscular suspension group at week 5 for the Beta variant (Figure 8).

Analysis of IgG antibodies revealed significantly higher levels in all treatment groups compared to the no-treatment groups across all four COVID-19 strains. No significant differences were observed between the intramuscular suspension and microneedle groups throughout the study period for all strains, except for the Omicron strain at week 2 and the Alpha strain in the terminal week (Figure 9). Two major subtypes of IgG, IgG1 and IgG2a, were analyzed to represent the Th2 and Th1 immune responses, respectively. IgG1-specific antibody levels showed significant differences between the vaccine microneedle treatment and no-treatment groups for all weeks against the Delta, Omicron, and Alpha strains; however, no significance was observed in the terminal week (week 7) for the Beta strain. When comparing the microneedle group to the intramuscular group, no significant differences were observed throughout the study period for all variants, except for the Delta variant (Figure 10). A similar trend was observed for IgG2a levels, where all antigen-specific responses, except for the Beta-specific antibody response, were significant in the microneedle group compared to the no-treatment group during the study period. When comparing intramuscular suspension responses to microneedle responses, no significance was observed across the study period for Omicron-, Delta-, and Beta-specific antibodies, except in week 2, when the intramuscular response was significant compared to microneedle for the Delta-specific antibody. For the Alpha-specific antibody, the microneedle group showed significance against the intramuscular group in week 2 and week 5 (Figure 11).

#### 3.4.2. Mucosal Antibody IgA Response to In Vivo Administration

IgA-specific secretory antibody responses were analyzed against Delta, Omicron, Alpha, and Beta strains of SARS-CoV-2 using ELISA. A significant difference was observed in the treatment vs. naïve group throughout the timeline of the study period for all four strains (Figure 12).

## 4. Discussion

Our heterologous vaccine, comprising two variants of SARS-CoV-2, namely, Delta and Omicron, was formulated for delivery via microneedles. The microneedle-based vaccine delivery method eliminates the risks associated with needle-stick injuries, allowing for self-administration and increasing accessibility and convenience. This needle-free approach eliminates the risk of cross-contamination among individuals. Moreover, this formulation technology is cost-effective and easily integrated into vaccine form, making it an attractive approach for mass production. Moreover, unlike intramuscular vaccines, microneedle vaccines can often be stored at room temperature, bypassing the need for costly cold chain logistics, thus improving vaccine distribution efficiency [29]. Microneedles, when inserted into the skin, temporarily disrupt the stratum corneum and penetrate the epidermis without reaching the pain receptors or nerve endings in the dermis [30]. The vaccine microparticles are then directly released into the dermis layer. The skin contains skin-associated lymphoid tissue (SALT), which activates both innate and adaptive immune responses by stimulating antigen-presenting cells, such as Langerhans cells and dendritic cells. These cells capture the antigen and transport it to the lymph nodes, triggering both cellular and humoral immune responses, including the activation of B lymphocytes [30,31]. While the transdermal route is pain-free and effective in generating immune responses, it can occasionally cause skin irritation. To mitigate this, we used hyaluronic acid (HA) in our formulation, a component of the skin’s extracellular matrix commonly derived from various animal sources for commercial use. HA is biocompatible, biodegradable, and non-immunogenic, with additional properties like wound healing and skin hydration. As a humectant, HA draws water into the skin, moisturizing the stratum corneum and enhancing permeability, ultimately aiding in the efficient delivery of vaccine microparticles across the dermis layer [32,33,34].

Microparticles provide antigen stability by encapsulating them in a polymeric matrix. The poly(lactic-co-glycolic acid) (PLGA) polymer used here is classified as safe by the Food and Drug Administration (FDA) and the European Medicines Agency (EMA), making it suitable for drug delivery and biomedical applications. The metabolites of PLGA, lactic acid, and glycolic acid, degrade easily under normal physiological conditions. According to the literature, encapsulating the antigen in PLGA provides long-term immune responses, as it acts as a controlled delivery system that prolongs antigen presentation to antigen-presenting cells (APCs) [35,36]. The extended release is associated with the release profile of the PLGA-based formulation, which has an initial rapid release followed by sustained release [37]. The results obtained from the in vitro Griess assay indicate that the microparticulate group showed significance compared to the no-treatment group, thus activating DCs in the process. This is primarily triggered by the particle size of PLGA-based microparticles, which have the ability to act as a depot, thus prolonging exposure to the DCs [38]. The particle size of our vaccine microparticulate formulation was in the range of 650–750 nm, which is another factor that influences the uptake of microparticles. Zeta potential is also one of the important parameters that reflects the electrostatic nature of the particles. A highly positive or negative zeta potential indicates the prevention of particle aggregation [38]. Our particles were in the range of −40 to −50 mV, which shows that they will be well suspended and evenly dispersed.

Qualitative analysis of the antigen was carried out using SDS-PAGE to analyze the molecular weight of the antigen after formulating the microparticles [39]. The results show that there was protein bond formation between 75 and 50 Kda in both the antigen suspension and the microparticulate form of the antigen. This could be because of the monomeric form of the nucleocapsid protein, which is present in higher concentration and is easily detected.

We tested the microparticulate vaccine in vitro to determine cell viability using the MTT assay. The cells in the treatment group demonstrated viability when compared to those in the positive control group, as evidenced by the significant conversion of tetrazolium to purple formazan crystals. This indicates that the formulations were non-cytotoxic to the dendritic cells [40,41]. To evaluate the immune-stimulatory effect, a necessary component of vaccine efficacy, we performed the Griess assay, in which we assessed and quantified the nitric oxide content released by dendritic cells when exposed to the vaccine microparticles. The results showed that the vaccine microparticles, along with the adjuvants group, showed significance when compared to the antigen in suspension form. This implies that the vaccine microparticles are more immunogenic than the suspension form. The production of nitric plays a key role in activating and differentiating T-cells, thus bridging innate and adaptive immunity [42,43]. Moreover, the in vitro results from the cytotoxicity assay confirm that cytotoxicity was not observed, even at the highest concentration (500 µg/mL) of the vaccine formulation, when compared to the positive control DMSO group. Thus, our vaccine formulations were non-cytotoxic to the cells.

In this study, we analyzed the production of different immunoglobulins found in mouse serum post-vaccination. Immunoglobulins are part of the humoral immune response formed after B-cell activation triggered by antigen entry. The antigen binds to the B-cell receptor on the surface, prompting B cells to differentiate into plasma cells that generate different types of immunoglobulins (Ig) [44]. We evaluated IgM levels in mouse serum after administering the vaccine. The levels peaked when serum was analyzed a week after the prime and booster doses of the vaccine, specifically, at week 2 and week 5, respectively. This confirms that the immune response was triggered in both the intramuscular suspension and microneedle groups, demonstrating that microneedle administration was as efficacious as traditional intramuscular administration. IgM is the first antibody generated when the immune system is activated. It subsequently undergoes seroconversion into IgG and IgA [45,46]. IgA is the second most abundant circulating neutralizing antibody involved in opsonization and pathogen removal. It is primarily secreted in mucosal areas, often at the point of entry of the pathogen into the immune system [47]. Alpha- and Beta-specific cross-reactivity was observed throughout the entire study period for IgA and IgM serum antibodies. Mutations in the spike protein, such as D614G, and receptor-binding domain (RBD) mutations like N501Y and L452R are common in all four strains [48]. Mutations can alter the structure of the protein that binds to antibodies, thus making it ineffective. As these mutations were common in all four strains, the proteins involved in antigen-antibody binding that formed were similar for these four strains. Thus, antibodies formed against Omicron and Delta also exhibit an affinity for the Alpha and Beta strains. However, some variant-specific mutations, such as E484K at the RBD in the Beta variant, can impact antibody binding to the spike protein [49,50]. This could explain the lower levels of IgM and IgA responses observed for the Beta strain.

We also analyzed IgG levels, along with the IgG1 and IgG2a subtypes. IgG levels spiked in the treatment groups and remained elevated until the terminal week (week 7). IgG antibodies are considered one of the important antibodies in vaccine studies. The presence of IgG in serum confirms that the immune system has identified antigens and triggered an adaptive immune response [51]. Significant levels of antibodies in cross-reactive strains could be due to the polyclonal nature of antibodies that target multiple parts of the pathogens, including non-mutated regions of the receptor-binding domain [52,53]. Our observation depicts significantly higher IgG1 levels compared to IgG2a, indicating a Th2-biased immune response. The production of IgG1- and IgG2a-specific antibodies is associated with the Th2 response and the Th1 response, respectively. A Th2 bias suggests the production of more antibodies by activating B cells [54]. This implies that the vaccine is capable of producing a robust humoral response. Moreover, when the whole inactivated virus of the Delta variant-based vaccine was administered at week 2 and the serum samples were analyzed for Omicron variant-specific antibodies, a significant immune response was obsereved in week 2 for the IgG and IgA antibodies. This also indicates the generation of a cross-reactive immune response against the latest strain, Omicron. A rise in these antibody levels was observed after booster immunization and remained high throughout. Furthermore, significant levels of mucosal IgA antibodies were observed in all strains throughout this study. Mucosal IgA plays an important role in preventing the virus from entering through respiratory mucosal surfaces, which is the entry point for the COVID virus [55]. As a result, elevated levels of mucosal IgA can be beneficial in preventing the binding of the virus and the spread of infection.

The Omicron-specific microneedle groups demonstrated equivalent responses to the intramuscular suspension vaccine groups throughout this study. Likewise, equivalent efficacies were observed between groups for Delta-specific IgG and IgA antibodies. However, the intramuscular group showed significantly higher levels compared to the microneedle group for the IgG1 and IgM antibodies. The Alpha-specific IgG, IgM, and IgG1 responses showed no significance between treatment groups in all weeks; however, a significant response was observed for IgG2a and IgM antibodies. Beta-specific IgG, IgG2a, and IgG responses showed no significance between treatment groups in all weeks; however, a significant response was observed for IgM and IgA antibodies. This shows that the immunogenicity of the intramuscular group was comparable to that of the microneedle group.

Overall, partial cross-reactivity was achieved against most of the strains of COVID-19. The Beta strain showed limited cross-reactivity due to mutations in the receptor-binding domain and antigenic sites, leading to the antigenic drift that makes these sites less recognizable by antibodies. Moreover, these mutations alter the structure and differentiate Beta from other variants like Omicron and Delta, thus reducing binding with the antibodies [48]. Alpha, on the other hand, has several mutations, but it is less divergent compared to the Delta and Omicron variants [56]. This is reflected in the cross-reactive response against Delta- and Omicron-specific antibodies. The lower levels of response in Alpha and Beta strains may also be because the antibodies obtained from mouse sera were tested separately against the subunit part, which is the spike glycoprotein RBD of the virus obtained from the Alpha and Beta strains.

Our heterologous vaccination strategy could play a significant role in developing a more broadly effective COVID-19 vaccine, which is needed to fight the emerging mutations of the virus. Our further studies will include investigating the cellular immune response, and we also plan to assess the effectiveness of neutralizing antibodies in order to confirm the blocking ability of the virus and provide protection.

## 5. Conclusions

In this proof-of-concept study, we successfully formulated an adjuvanted microparticulate vaccine loaded in microneedles. Characterization of the microparticles confirmed even distribution, zeta potential, encapsulation efficacy, and morphology. In vitro assays confirmed the non-cytotoxicity and immunogenicity of the microparticles. Furthermore, in vivo assays demonstrated that the vaccinated groups effectively induced significant antibody levels in mice. Lastly, this microparticulate heterologous vaccine produced cross-reactive antibody responses to the other variants of SARS-CoV-2.

## Figures and Tables

**Figure 1 vaccines-13-00380-f001:**
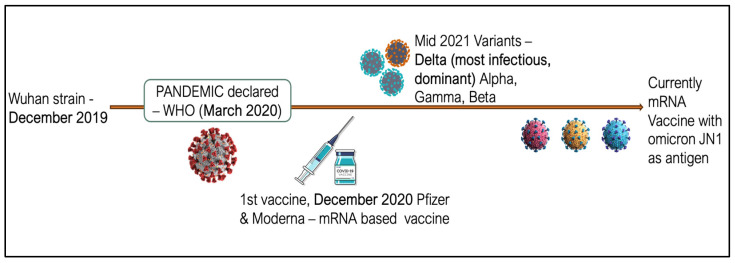
Timeline of virus mutations and vaccine development.

**Figure 2 vaccines-13-00380-f002:**
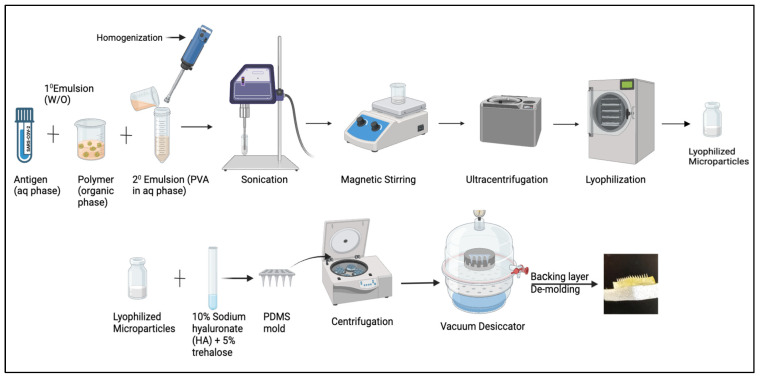
Formulation process for microparticulate microneedle vaccine.

**Figure 3 vaccines-13-00380-f003:**
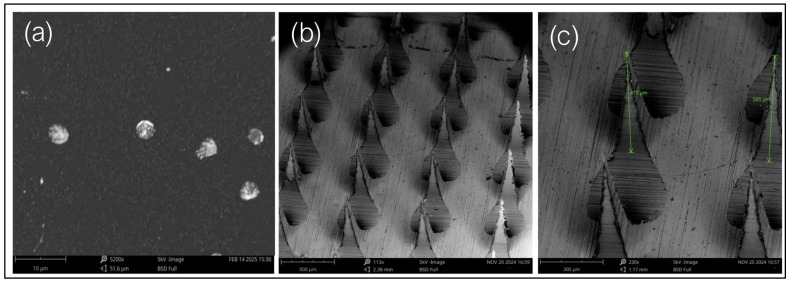
(**a**) Scanning electron microscopy of microparticle images at a magnification of 5200×, (**b**) scanning electron microscopy of microneedle images at a magnification of 113×, and (**c**) scanning electron microscopic images of microneedles representing an average height of 490 µm, comprising vaccine microparticles that are imaged at a magnification of 230×.

**Figure 4 vaccines-13-00380-f004:**
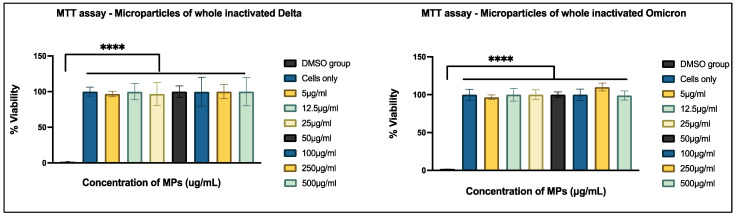
Cytotoxicity assessment of microparticles. Cells were seeded at a density of 1 × 10⁴ cells per well and exposed to treatment groups for 24 h. The treatment groups included DMSO (positive control), untreated cells (negative control), and microparticulate vaccines at varying concentrations: 500 µg/mL, 250 µg/mL, 100 µg/mL, 50 µg/mL, 25 µg/mL, 12.5 µg/mL, and 5 µg/mL. Data are presented as mean ± SEM and analyzed using one-way ANOVA, followed by Tukey’s post hoc multiple comparison test (**** *p* < 0.0001).

**Figure 5 vaccines-13-00380-f005:**
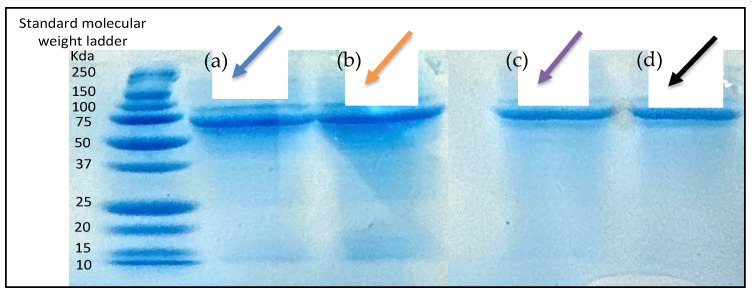
SDS page analysis. Column (**a**) blue arrow represents bands formed by Delta suspension; Column (**b**) orange arrow represents bands formed by Omicron suspension. Column (**c**) purple arrow represents bands formed by Delta microparticles; Column (**d**) black arrow represents bands formed by Omicron microparticles.

**Figure 6 vaccines-13-00380-f006:**
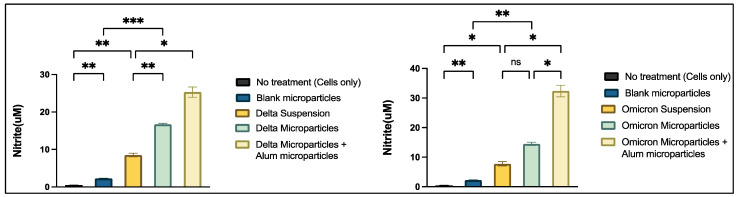
Nitrite release determination using Griess assay. Cells were seeded at a density of 1 × 10⁴ cells per well and exposed to various treatment groups for 24 h. The treatment groups included untreated cells (no treatment), blank microparticles (2 µg), whole inactivated virus Delta suspension (2 µg), microparticulate vaccine with whole inactivated virus Delta variant (2 µg), microparticulate vaccine with WIV Delta variant + adjuvant microparticles (2 µg), whole inactivated virus Omicron suspension (2 µg), microparticulate vaccine with whole inactivated virus Omicron variant (2 µg), and microparticulate vaccine with whole inactivated virus Omicron variant + adjuvant microparticles (2 µg). Following incubation, supernatants were collected and analyzed for nitrite release using the Griess assay. Data are presented as mean ± SEM and were analyzed using one-way ANOVA, followed by Tukey’s post hoc multiple comparison test (* *p* ≤ 0.05, ** *p* ≤ 0.01, *** *p* ≤ 0.001).

**Figure 7 vaccines-13-00380-f007:**
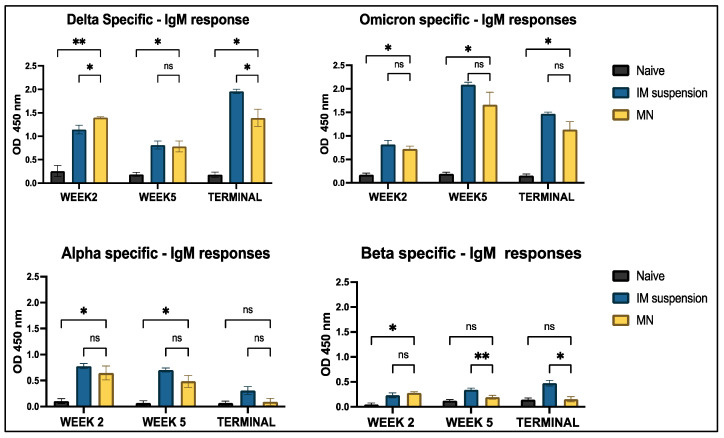
Antigen-specific antibody levels in serum for IgM were measured in vaccinated and unvaccinated mice. The serum was analyzed using ELISA at a dilution of 1:50, and cross-reactive responses against Alpha and Beta strains were measured. Data are expressed as mean ± SEM and were analyzed using two-way ANOVA followed by post hoc Tukey’s multiple comparison test. * *p* < 0.05, ** *p* < 0.01, ns—non-significant.

**Figure 8 vaccines-13-00380-f008:**
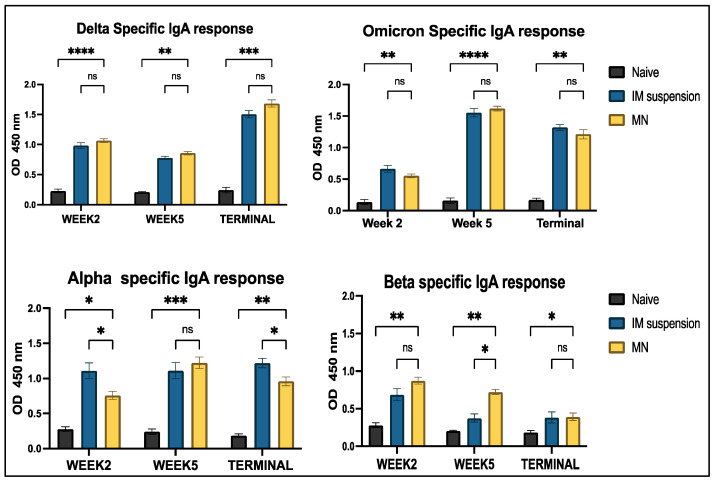
Antigen-specific antibody levels in serum for IgA were measured in vaccinated and unvaccinated mice. Serum was analyzed using ELISA at a dilution of 1:50, and cross-reactive responses against Alpha and Beta strains were measured. Data are expressed as mean ± SEM and were analyzed using two-way ANOVA followed by post hoc Tukey’s multiple comparison test. * *p* ≤ 0.05, ** *p* < 0.01, *** *p* < 0.001, **** *p* < 0.0001, ns—non-significant.

**Figure 9 vaccines-13-00380-f009:**
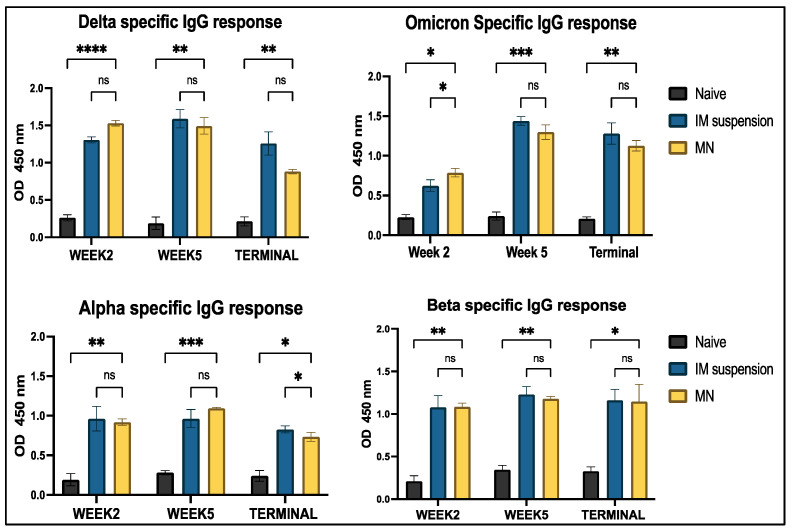
Antigen-specific IgG antibody levels in serum were measured in vaccinated and unvaccinated mice. Serum samples were analyzed using ELISA at a 1:50 dilution, and cross-reactive responses against Alpha and Beta strains were assessed. Data are presented as mean ± SEM and were analyzed using two-way ANOVA, followed by Tukey’s post hoc multiple comparison test (* *p* ≤ 0.05, ** *p* < 0.01, *** *p* < 0.001, , **** *p* < 0.0001 ns—non-significant).

**Figure 10 vaccines-13-00380-f010:**
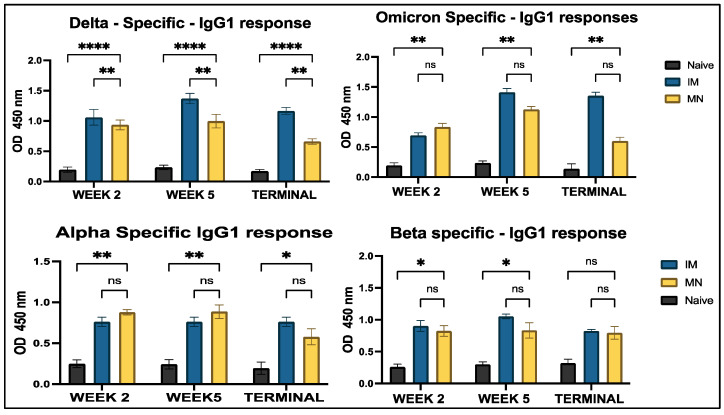
Antigen-specific antibody levels in serum for IgG1 were measured in vaccinated and unvaccinated mice. The serum was analyzed using ELISA at a dilution of 1:50. Cross-reactive responses against Alpha and Beta strains were measured. Data are expressed as mean ± SEM and were analyzed using two-way ANOVA followed by post hoc Tukey’s multiple comparison test.(* *p* ≤ 0.05, ** *p* < 0.01, **** *p* < 0.0001, ns—non-significant).

**Figure 11 vaccines-13-00380-f011:**
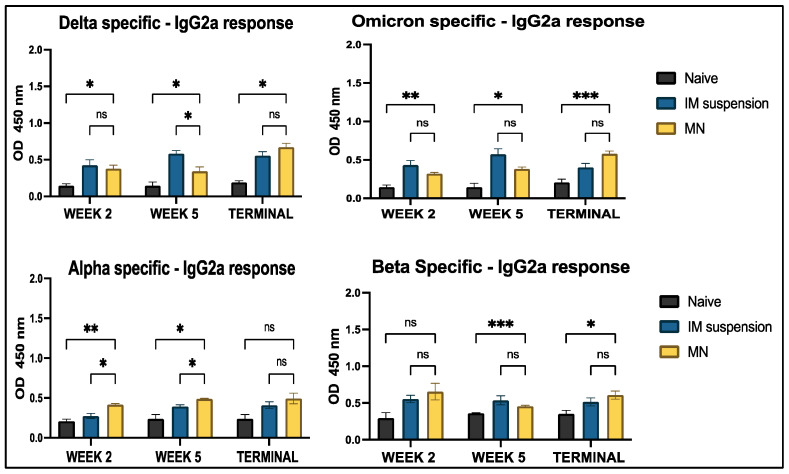
Antigen-specific antibody levels in serum for IgG2a were measured in vaccinated and unvaccinated mice. The serum was analyzed using ELISA at a dilution of 1:50, and cross-reactive responses against Alpha and Beta strains were measured. Data are expressed as mean ± SEM and were analyzed using two-way ANOVA followed by post hoc Tukey’s multiple comparison test. (* *p* ≤ 0.05, ** *p* < 0.01, *** *p* < 0.001, , ns—non-significant).

**Figure 12 vaccines-13-00380-f012:**
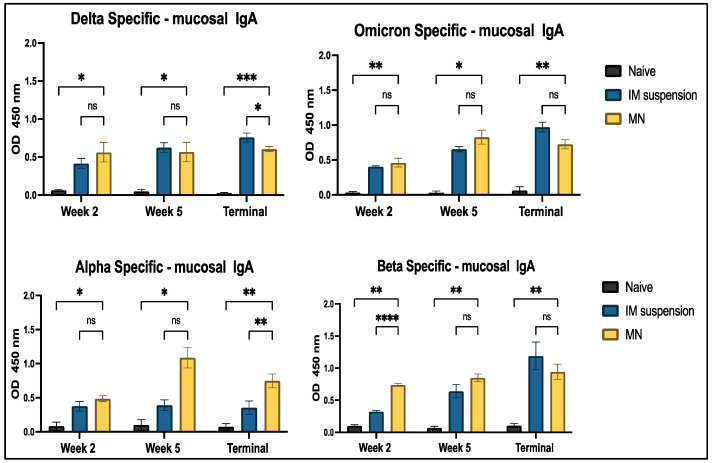
Antigen-specific antibody levels in vaginal samples for IgA were measured in vaccinated and unvaccinated mice. The serum was analyzed using ELISA at a dilution of 1:50, and cross-reactive responses against Alpha and Beta strains were measured. Data are expressed as mean ± SEM and were analyzed using two-way ANOVA followed by post hoc Tukey’s multiple comparison test. (* *p* ≤ 0.05,** *p* < 0.01, *** *p* < 0.001, **** *p* < 0.0001, ns—non-significant).

**Table 1 vaccines-13-00380-t001:** Description of vaccination groups.

Study Groups	Route of Administration	Dose of Whole Inactivated Virus and Adjuvant
Naïve (No Treatment)	NA	NA
Intramuscular Suspension	Intramuscular Injection	Prime Dose: 40 µg WIV Delta Antigen + 30 µg Alum Booster Dose: 40 µg WIV Omicron Antigen + 30 µg Alum
Microparticulate Microneedle	Intradermal Microneedle	PLGA-Based Microparticles in Dissolving Microneedles Prime Dose: 40 µg WIV Delta Antigen + 30 µg Alum Booster Dose: 40 µg WIV Omicron Antigen + 30 µg Alum

**Table 2 vaccines-13-00380-t002:** Microparticle characterization.

	Whole Inactivated Virus —Delta Variant	Whole Inactivated Virus —Omicron Variant
Average particle size	692.7 nm ± 38.7	731 nm ± 21.3
Particle count	1197 ± 49.16	1227.3 ± 28.9
Zeta potential	−52.9 ± 19.9	−43 mV ± 15.2
Encapsulation efficacy	86.5 ± 0.9	89.12 ± 0.77

## Data Availability

The data presented in this study are available upon request from the corresponding author.

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
