# Peer review of "Microneedle Delivery of Heterologous Microparticulate COVID-19 Vaccine Induces Cross Strain Specific Antibody Levels in Mice"

_vaccines, 2025, doi:10.3390/vaccines13040380_

Round 1
Reviewer 1 Report
Comments and Suggestions for Authors
This proof-of-concept study aimed to assess the effectiveness of microneedle assisted delivery of a heterologous microparticulate vaccine against the SARS-CoV-2 virus. This research used a heterologous vaccine strategy approach using inactivated viruses from Delta and Omicron variants, delivered via transdermal microneedle. The research team encapsulated viral antigens in PLGA microparticles with alum adjuvant. Experimental results demonstrated that microneedle microparticulate based heterologous vaccine showed cross-reactivity within the strains. However, there are some points that the authors should address that can improve the manuscript.
- The manuscript explicitly lists the doses of antigens (Delta and Omicron) and adjuvants but does not provide detailed justification for these doses. It is recommended to supplement the rationale behind the dose selection.
- In the 2.5 In Vivo Administration section, the manuscript describes specific procedures for mouse experiments, including vaccination and serum collection. Since these experiments involve the use of animals, it is recommended to add an ethical statement.
- Please supplement detailed information about the experimental design, such as the mouse grouping (whether randomization was performed), animal number selection, and sample size calculation.
- The manuscript does not provide detailed descriptions of the housing conditions for mice or the injection sites for the two administration methods (intramuscular and microneedle). It is recommended to supplement this information in the 2.5 In Vivo Administration section to enhance the reproducibility of the experiments and the reliability of the results.
- It is recommended to provide specific information about the syringes and microneedles used in the Intramuscular Suspension and Microparticulate Microneedle groups. This should include basic details about the devices, such as flow rate, the use of dye to measure the penetration depth of the injected liquid, and the contact area of the syringe.
- The formatting of references is inconsistent in some parts of the manuscript. It is recommended to unify the reference format and supplement missing DOI information where applicable.
- There are some errors in the manuscript details. For example, the title "2.3.2 Scanning Electron Microscopy" is repeated, and "Results 3" appears in line 164. It is recommended to carefully review and correct these issues.
- Please verify that the data in the tables are consistent with the text descriptions, such as in Section 3.1: "The average particle size for Delta MPs and Omicron MPs was 731± 21.3 nm and 692.7nm ± 38.7, respectively." This sentence contradicts the data in the table.
- The article only provides statistical significance analysis (e.g., p-values). It is recommended to add a report on effect size.
- The discussion section could be expanded to include a comparison between the microneedle delivery system and other existing delivery methods (such as intramuscular injection, intranasal vaccines, etc.).
- In the discussion section, the cross-protection data of the vaccine against the latest variant strains (for example BA.4/5) should be supplemented to enhance the practical application value.
Author Response
This proof-of-concept study aimed to assess the effectiveness of microneedle assisted delivery of a heterologous microparticulate vaccine against the SARS-CoV-2 virus. This research used a heterologous vaccine strategy approach using inactivated viruses from Delta and Omicron variants, delivered via transdermal microneedle. The research team encapsulated viral antigens in PLGA microparticles with alum adjuvant. Experimental results demonstrated that microneedle microparticulate based heterologous vaccine showed cross-reactivity within the strains. However, there are some points that the authors should address that can improve the manuscript.
Specific Comments:
- The manuscript explicitly lists the doses of antigens (Delta and Omicron) and adjuvants but does not provide detailed justification for these doses. It is recommended to supplement the rationale behind the dose selection.
Response – We thank the reviewer for their comment. The studies conducted in past utilized whole inactivated approach of SARS-CoV-2. We had used 20ug of antigen along with one prime and two booster immunizations to generate immune response. However, the immune response generated had low antibody titers even when 2 boosters of whole inactivated virus was used in these previous studies. (references below). Based on these previous studies, in this ‘proof-of-concept’ study we used 40ug of antigen dose with only one booster dosing regimen.
(Vijayanand, S., Patil, S., Menon, I., Braz Gomes, K., Kale, A., Bagwe, P., Uddin, M. N., Zughaier, S. M., & D’Souza, M. J. (2023). An Adjuvanted Inactivated SARS-CoV-2 Microparticulate Vaccine Delivered Using Microneedles Induces a Robust Immune Response in Vaccinated Mice. Pharmaceutics, 15(3), 895. https://doi.org/10.3390/pharmaceutics15030895)
(Vijayanand S, Patil S, Joshi D, Menon I, Braz Gomes K, Kale A, Bagwe P, Yacoub S, Uddin MN, D’Souza MJ. Microneedle Delivery of an Adjuvanted Microparticulate Vaccine Induces High Antibody Levels in Mice Vaccinated against Coronavirus. Vaccines. 2022; 10(9):1491. https://doi.org/10.3390/vaccines10091491)
- In the 2.5 In Vivo Administration section, the manuscript describes specific procedures for mouse experiments, including vaccination and serum collection. Since these experiments involve the use of animals, it is recommended to add an ethical statement.
Response – We thank the reviewer for the appropriate comment. The testing was carried out as per protocol. Animal procedures were designed to minimize animal suffering and distress. Study adhered to the ethical principles of the 3Rs (Replacement, Reduction, and Refinement). This statement is now added in section 2.5 of the revised manuscript in the line 251- 254.
3. Please supplement detailed information about the experimental design, such as the mouse grouping (whether randomization was performed), animal number selection, and sample size calculation.
Response – Thank you for the comment. To determine the sample size for in-vivo studies, the following equation was used: N = 1+2C(s/d); where ‘N’ is the sample size, ‘s’ is the standard deviation calculated based on recent vaccine studies, which was 1, ‘d’ is the difference between the vaccinated and the non-vaccinated groups. C is a constant, which is calculated using α as 0.05 (type I error) and β (type II error) as 0.20, and with the power of 80%. Based on these calculations, the sample size was 6.
4. The manuscript does not provide detailed descriptions of the housing conditions for mice or the injection sites for the two administration methods (intramuscular and microneedle). It is recommended to supplement this information in the 2.5 In Vivo Administration section to enhance the reproducibility of the experiments and the reliability of the results.
Response – We thank the reviewers for the important observation. Mice were housed in a controlled environment with food and water and they were monitored daily for signs of distress. This statement is now added in manuscript in the line 254-255.
Intramuscular administration was performed by administrating the dose into thigh muscle using a 26-gauge needle after restraining the animal. For administrating microneedles dorsal side of the mice was used and shaved 24hours prior administration so that hair interference doesn’t affect dosing. Next day, mice were placed in anesthesia chamber to restrict the moment and access the administration site. The skin was pulled back on a small wooden block and using a microneedle spring applicator, effective administration was performed by holding the applicator over the skin.
This information is now mentioned in the line 257 to 263 of the revised manuscript.
5. It is recommended to provide specific information about the syringes and microneedles used in the Intramuscular Suspension and Microparticulate Microneedle groups. This should include basic details about the devices, such as flow rate, the use of dye to measure the penetration depth of the injected liquid, and the contact area of the syringe.
Response – We thank the reviewer for this very thoughtful suggestion. This statement related to the administration device and use of needle is added in the revised manuscript, line 257-258 and 261 to 263.
In the previous studies we had used same protocol for formulation and delivery of the microneedles across the skin. This manuscript mentions the depth of penetration of microneedles. The dye used is Methylene blue. (Braz Gomes, K.; D’Souza, B.; Vijayanand, S.; Menon, I.; D’Souza, M.J. A Dual-Delivery Platform for Vaccination Using Antigen-Loaded Nanoparticles in Dissolving Microneedles. International Journal of Pharmaceutics 2022, 613, 121393, doi:10.1016/j.ijpharm.2021.121393.)
6. The formatting of references is inconsistent in some parts of the manuscript. It is recommended to unify the reference format and supplement missing DOI information where applicable.
Response – Thank you reviewer for that observation. We have now added DOI for reference 33 and 35. Unfortunately, we cannot add the DOI in the reference 1, 2, 7, 39, 44 since these are either website pages or book articles that are without DOI information.
- There are some errors in the manuscript details. For example, the title "2.3.2 Scanning Electron Microscopy" is repeated, and "Results 3" appears in line 164. It is recommended to carefully review and correct these issues.
Response – Thank you for the comment. These repetitive Titles have now been removed from the current uploaded manuscript.
8. Please verify that the data in the tables are consistent with the text descriptions, such as in Section 3.1: "The average particle size for Delta MPs and Omicron MPs was 731± 21.3 nm and 692.7nm ± 38.7, respectively." This sentence contradicts the data in the table.
Response – We thank the authors for noticing this error. We have now corrected the statement in section 3.1. The currently uploaded manuscript reflects the corrected statement is on the line 308 and 309 of the revised manuscript.
9. The article only provides statistical significance analysis (e.g., p-values). It is recommended to add a report on effect size.
Response – We thank the reviewer for the comment. We have now used two-way ANOVA as statistical test in ELISA related data, as we have different weeks and different study groups so the data presented is represented in form of p-values.
- The discussion section could be expanded to include a comparison between the microneedle delivery system and other existing delivery methods (such as intramuscular injection, intranasal vaccines, etc.).
Response – We thank the reviewer for the encouraging critique and appreciate constructive feedback. We have included this in the introduction part on the line 84 to 92 ‘Conventionally, vaccines are delivered using intramuscular delivery, however, need of trained physicians, lack of patient compliance and potential risks of hematoma, had led to need for an alternative delivery strategy. As the research progresses in developing non-invasive vaccination route, intra-nasal and intra-dermal delivery systems are explored. Vaccine delivered intranasally has a major drawback with effective immune activation due to the limited ability of the vaccine to cross mucosal barrier. On the other hand, microneedles administration delivered through skin is devoid of this limitation and would rather help eliminate vaccine hesitancy among individuals due to needle phobia because of the hypodermal needles.
Moreover, discussion part also provides details from line 469 to 476 of revised manuscript.
‘Microneedle based vaccine delivery method eliminate the risks associated with needle-stick injuries, allowing for self-administration, increasing accessibility and convenience. This needle- free approach eliminates the risk of cross contamination among individuals. Moreover, this formulation technology is cost effective and easily integrated into vaccine form, making it an attractive approach for mass production. Moreover, unlike intramuscular vaccines, these microneedle vaccine formulations can be stored at room temperature, bypassing the need for costly cold-chain logistics, thus improving vaccine distribution efficiency’.
- In the discussion section, the cross-protection data of the vaccine against the latest variant strains (for example BA.4/5) should be supplemented to enhance the practical application value.
Response – We thank the reviewer for the encouraging critique and appreciate constructive feedback. We have tested cross reactive response against the latest variant that is omicron strain. Cross Reactivity was seen in week 2 when the animals were administered with whole inactivated virus with delta variant. After booster immunization, the responses for omicron specific antibodies increased and remained significantly high throughout the study. We have now mentioned this in line 570-576.
Reviewer 2 Report
Comments and Suggestions for Authors
The following paper is a study that observed cross-reactive immunity using a whole-inactivated virus vaccine in PLGA nanoparticles within microneedles. Overall, while the study is intriguing, there are areas that need to be further strengthened. The questions are as follows.
- Differentiation from Existing Studies
- Please refer to existing studies that have utilized PLGA and microneedles for vaccine delivery, and, with appropriate reference reinforcement, emphasize in the introduction how the current study differentiates itself. - Specification of Alum Delivery Form
- Clearly state how Alum was delivered in the experiment.
- Was it incorporated within PLGA, within HA microneedles, or administered separately? Please clarify. - Alum Administration Site and Dosage
- If Alum was administered separately, specify whether the administration site is the same as that of the microneedles and indicate the dosage used. - Clarification on Table 1 Composition
- In Table 1 on page 8, please explain why there is no group without Alum in both the IM and microneedle groups. - Rationale for Selecting the Alum Amount
- Provide a detailed explanation for the choice of 30 μg of Alum. - Question Regarding Zeta Potential Values
- In Table 2, the zeta potential of the micro(nano) particles is around –50 mV. Is this value consistent with those reported in other studies? If so, discuss the reasons behind such a significant value. - Correction of Unit Error
- The unit “uM” indicated in line 295 is incorrect; please correct it to the appropriate unit. - Revision Request for Figure 4
- Please reorganize and modify Figure 4 for improved clarity and presentation, and move the ladder from the center to the left edge. - Clarification on “Alum Microparticle” in Figure 5
- In Figure 5, the cell experiment is labeled “Alum microparticle.” Please clarify what this label signifies and provide further details. - Analysis of Drug Release Patterns
- According to the Griess assay, aside from the case with Alum, there appears to be little difference between the suspension and particle groups. Considering that PLGA releases the vaccine over an extended period, please describe the drug release patterns for Delta microparticles and Omicron microparticles. - Discussion on Dendritic Cell Activation
- Based on existing studies and validated evidence that PLGA microparticles activate dendritic cells, please expand the manuscript’s discussion to include these findings. - Specification of Contribution to Release Rate
- In microneedle delivery, both the release rate contributed by the microparticles and that attributed to the HA-based microneedles play roles. Please clarify how these two contributions are considered in the current study. - Animal Experiment Results for Alum-free Groups
- Indicate whether there are animal experiment results for the suspension and microparticle groups without Alum.
Author Response
The following paper is a study that observed cross-reactive immunity using a whole-inactivated virus vaccine in PLGA nanoparticles within microneedles. Overall, while the study is intriguing, there are areas that need to be further strengthened. The questions are as follows.
Specific Comments:
- Differentiation from Existing Studies
- Please refer to existing studies that have utilized PLGA and microneedles for vaccine delivery, and, with appropriate reference reinforcement, emphasize in the introduction how the current study differentiates itself.
Response – We thank the reviewer for the encouraging critique and appreciate constructive feedback. In our previous studies, microparticulate vaccine delivered by microneedle has shown to generate excellent immune responses for other vaccines such as Influenza, Gonorrhea, Respiratory Syncytial Virus, Zika, Coronavirus. Research have used Poly (lactic-glycolic acid) (PLGA) as a polymer to formulate microparticles. The studies confirm that, when polymers like PLGA is used as matrix it provides stability and prevent antigen degradation at elevated temperatures. Moreover, shape and size of PLGA based microparticles govern the sustained delivery profile of the antigens. Microparticles with a size between 100nm to 1000nm are efficiently up taken by antigen presenting cells (APCs) and produce a strong immune response. Furthermore, being a bio-degradable polymer, these vaccine formulations are non-toxic. Thus, PLGA polymer was used to encapsulate the SARS-CoV-2 antigen in this study. We have mentioned this in the introduction paragraph on line 93-103 and referenced it.
In this manuscript, we also examined the delivery of heterologous PLGA based microparticulate vaccine against the SARS-CoV-2 virus using the Delta and Omicron antigens in combination. This vaccine strategy can produce broader immune responses as different variant are combined in this protocol. Moreover, in this study we also examined cross reactive responses to the Alpha and Beta strains. This is now mentioned in line 76-79 of the uploaded manuscript.
- Specification of Alum Delivery Form
- Clearly state how Alum was delivered in the experiment.
- Was it incorporated within PLGA, within HA microneedles, or administered separately? Please clarify.
Response – We thank the reviewer for this comment. Alum microparticles were formulated using double emulsion method, wherein alum was encapsulated into the PLGA polymer and microparticles were formulated. The process of formulating alum microparticles is similar to the method used for the formulation of the vaccine antigen microparticles. The vaccine microparticles along with the alum microparticles were then incorporated into HA along with trehalose and formulated into microneedles. We have now mentioned this in the revised manuscript in lines 152-156.
- Alum Administration Site and Dosage
- If Alum was administered separately, specify whether the administration site is the same as that of the microneedles and indicate the dosage used.
Response – We formulated Microneedles comprising of alum microparticles along with the vaccine microparticles in the same microneedle array. This is now mentioned in the lines 155-156 of the revised manuscript.
Administration of microneedles was performed on dorsal side of mice. We have mentioned this information in line 290.
4. Clarification on Table 1 Composition
- In Table 1 on page 8, please explain why there is no group without Alum in both the IM and microneedle groups.
Response – We thank the reviewer for making that observation. In the past your lab has performed animal-based experiments to analyze the immune responses for vaccine antigen along with adjuvant and vaccine antigen without adjuvant. Results showed that, groups with Alum (adjuvant) showed higher significance as compared to groups without alum in two previous studies as referenced below. As the data was consistent, in an attempt to reduce the animal usage we did not included intramuscular or microneedle group without Alum in this study.
(Kale, A.; Joshi, D.; Menon, I.; Bagwe, P.; Patil, S.; Vijayanand, S.; Braz Gomes, K.; D’Souza, M. Novel Microparticulate Zika Vaccine Induces a Significant Immune Response in a Preclinical Murine Model after Intramuscular Administration. International Journal of Pharmaceutics 2022, 624, 121975, doi:10.1016/j.ijpharm.2022.121975.)
(Patil, S.; Vijayanand, S.; Menon, I.; Gomes, K.B.; Kale, A.; Bagwe, P.; Yacoub, S.; Uddin, M.N.; D’Souza, M.J. Adjuvanted-SARS-CoV-2 Spike Protein-Based Microparticulate Vaccine Delivered by Dissolving Microneedles Induces Humoral, Mucosal, and Cellular Immune Responses in Mice. Pharmaceuticals 2023, 16, 1131, doi:10.3390/ph16081131)
5. Rationale for Selecting the Alum Amount
- Provide a detailed explanation for the choice of 30 μg of Alum.
Response – Previous published COVID19 vaccine study used Alum was used as an adjuvant. This paper focuses on COVID-19 study that also used 30μg of Alum. Based on the previous literature in this proof-of-concept study, we thus have used same concentration of Alum ( Patil, S.; Vijayanand, S.; Menon, I.; Gomes, K.B.; Kale, A.; Bagwe, P.; Yacoub, S.; Uddin, M.N.; D’Souza, M.J. Adjuvanted-SARS-CoV-2 Spike Protein-Based Microparticulate Vaccine Delivered by Dissolving Microneedles Induces Humoral, Mucosal, and Cellular Immune Responses in Mice. Pharmaceuticals 2023, 16, 1131, doi:10.3390/ph16081131.)
6. Question Regarding Zeta Potential Values
- In Table 2, the zeta potential of the micro(nano) particles is around –50 mV. Is this value consistent with those reported in other studies? If so, discuss the reasons behind such a significant value.
Response – We thank the reviewer for making that important observation. The zeta potential for the PLGA based microparticles have high negative zeta potential due to the presence of carboxyl (-COOH) functional group that are present in the PLGA polymer. Particles having zeta potential above
-20mV have the characteristics of repelling each other, thereby enhancing the stability by preventing aggregation. Studies have shown with a optimization of formulation zeta potential can cross -60mV indicating higher stability and strong electrostatic repulsions.
(Stromberg, Z.R.; Lisa Phipps, M.; Magurudeniya, H.D.; Pedersen, C.A.; Rajale, T.; Sheehan, C.J.; Courtney, S.J.; Bradfute, S.B.; Hraber, P.; Rush, M.N.; et al. Formulation of Stabilizer-Free, Nontoxic PLGA and Elastin-PLGA Nanoparticle Delivery Systems. International Journal of Pharmaceutics 2021, 597, 120340, doi:10.1016/j.ijpharm.2021.120340.)
- Correction of Unit Error
- The unit “uM” indicated in line 295 is incorrect; please correct it to the appropriate unit.
Response – We thank the reviewer for the constructive feedback. We have replaced uM with μm. The changes will be reflected on the line 317 in the new version of revised manuscript.
- Revision Request for Figure 4
- Please reorganize and modify Figure 4 for improved clarity and presentation, and move the ladder from the center to the left edge.
Response – Thank you for the comment. We have now modified figure 4 and included in the revised manuscript.
- Clarification on “Alum Microparticle” in Figure 5
- In Figure 5, the cell experiment is labeled “Alum microparticle.” Please clarify what this label signifies and provide further details.
Response – Figure 5 comprises of Analysis after performing Griess Assay. In this assay there is no separate Alum microparticle group. We had groups like Delta Microparticles + Alum Microparticles and Omicron Microparticle + Alum microparticles because we wanted to test the immune-stimulatory response of the combination of Antigen microparticles along with adjuvant microparticles within the Dendritic cells.
- Analysis of Drug Release Patterns
- According to the Griess assay, aside from the case with Alum, there appears to be little difference between the suspension and particle groups. Considering that PLGA releases the vaccine over an extended period, please describe the drug release patterns for Delta microparticles and Omicron microparticles.
Response – Thank you for this comment. In the past we have carried out done an extensive characterization of the release profile of PLGA based Microparticles with an half-life of release of 24hours, followed by a sustained release profile. We expected a similar release profile for this cargo because we have used the same polymer and formulation method for the encapsulation process. (Braz Gomes, K., Vijayanand, S., Bagwe, P., Menon, I., Kale, A., Patil, S., Kang, S. M., Uddin, M. N., & D'Souza, M. J. (2023). Vaccine-Induced Immunity Elicited by Microneedle Delivery of Influenza Ectodomain Matrix Protein 2 Virus-like Particle (M2e VLP)-Loaded PLGA Nanoparticles. International journal of molecular sciences, 24(13), 10612. https://doi.org/10.3390/ijms241310612)
In this experiment we have exposed the cells with microparticles for 24hours which means that due to the sustained release profile dendritic cells (DCs) will be stimulated for extended period of time as compared to the antigen suspension group where DCs are stimulated at the time of exposure. Therefore, we see significance in the Antigen + adjuvant MP treatment group when compared to the Antigen suspension group.
- Discussion on Dendritic Cell Activation
- Based on existing studies and validated evidence that PLGA microparticles activate dendritic cells, please expand the manuscript’s discussion to include these findings.
Response - We thank the reviewer for this comment. Immune response is triggered after activation of antigen presenting cells on exposure to foreign substances, including vaccine antigens. The results obtained from in-vitro Griess assay, indicate the microparticulate group showed significance across DC cells only group, thus activating DCs in the process. This is primarily triggered by the particle size of PLGA based microparticles, and its ability to act as depot thus prolonging the exposure to the DCs. This is indicated in the line 501 to 505 of revised manuscript.
- Specification of Contribution to Release Rate
- In microneedle delivery, both the release rate contributed by the microparticles and that attributed to the HA-based microneedles play roles. Please clarify how these two contributions are considered in the current study
Response – Thank you reviewers for the comment. We have formulated microparticle based microneedle vaccine. Microneedles ensures effective delivery of microparticles across epidermis layer of the skin. Microparticles have a sustained release profile due to use of PLGA as a polymer. Microparticles were added to the HA and trehalose mixture to form microneedles. HA based microneedles when administered onto skin has a rapid release profile, releasing microparticles immediately. The overall release of the antigen therefore is only controlled by the release pattern from PLGA polymer. We have mentioned this line in the discussion part of the manuscript on the line 499 to 501.
13. Animal Experiment Results for Alum-free Groups
- Indicate whether there are animal experiment results for the suspension and microparticle groups without Alum.
Response – In this study we have not included any animal experiments with just the Alum as an adjuvant group. In past our lab has performed animal-based experiments using different adjuvants in the animal based model using Neisseria meningitidis as a model antigen. The results show that Alum very significantly enhanced antigenicity and the antigen presentation in dendritic cells. (Gala, R.P.; D’Souza, M.; Zughaier, S.M. Evaluation of Various Adjuvant Nanoparticulate Formulations for Meningococcal Capsular Polysaccharide-Based Vaccine. Vaccine 2016, 34, 3260–3267, doi:10.1016/j.vaccine.2016.05.010.)
Reviewer 3 Report
Comments and Suggestions for Authors
The authors formulated microparticulated heat-inactivated SARS-CoV-2 virus vaccines using PLGA and evaluated them in mice by immunizing IM or MN administrations. MN technologies have gained popularity and attention in both academic and industry sectors. Although interesting, I’m not sure how the authors differentiated anti-SARS-CoV-2 antibodies among the 4 variants. For example, if the authors immunized mice twice with Delta vaccine, the sera don’t recognize other variants at all? Or is Delta prime / Omicron boost better than two immunizations with a Delta and Omicron mixture vaccine? Furthermore, antibody levels look quite low after two immunizations with 40µg of antigen with Alum adjuvant. I think the authors should measure neutralizing antibody against each variant to show true cross-reactivity. Specific comments follow.
Major points:
- Lines 164-166: Please delete these lines.
- Line 215 & 217: Please indicate the concentration of MTT and incubation temperature so that the readers can follow your method.
- Line 241: Please indicate number and sex of mice used.
- Line 260: Please make it clear about the ELISA antigens. Were they whole viruses?
- Lines 263-264: Please indicate the diluent for BSA.
- Figure 4: Please explain why only few bands are visible despite they are whole viruses. Can the authors include original viruses and perform western blot to confirm which band is which protein?
- Figure 5: Please show all the significant differences among groups as current graph does not show the difference between MP and MP+Alum groups.
Minor points:
- Line 14: “SARS-COV-2” should be “SARS-CoV-2”.
- Please use proper unit for microliter “µl” but not “ul” throughput the manuscript.
- Line 178: Please use consistent description throughput the manuscript. “mL” should be “ml”.
- Line 212: “ml” is missing after “500ug/”.
- Line 231: Please delete “a plate reader using” as this is a duplication.
- Line 302: “In-vivo” should be “In-vitro”.
- Figure 4: Please change color of the arrows for (c) and (d).
- Line 388: “1 and 50” should be “1:50”.
- Figure 9: Please change “Week” to “WEEK” and “Terminal” to “TERMINAL” for consistency.
Author Response
The authors formulated microparticulated heat-inactivated SARS-CoV-2 virus vaccines using PLGA and evaluated them in mice by immunizing IM or MN administrations. MN technologies have gained popularity and attention in both academic and industry sectors. Although interesting, I’m not sure how the authors differentiated anti-SARS-CoV-2 antibodies among the 4 variants. For example, if the authors immunized mice twice with Delta vaccine, the sera don’t recognize other variants at all? Or is Delta prime / Omicron boost better than two immunizations with a Delta and Omicron mixture vaccine? Furthermore, antibody levels look quite low after two immunizations with 40µg of antigen with Alum adjuvant. I think the authors should measure neutralizing antibody against each variant to show true cross-reactivity. Specific comments follow.
Response: We thank the reviewer for the encouraging critics and appreciate constructive feedback.
In this study, we have used a heterologous strategy of administering microneedle microparticulate vaccine to mice. We have administered whole inactivated virus of Delta variant at prime dosing and whole inactivated virus of Omicron at booster dosing. We have mentioned this in detailed in table 1 of the manuscript. Mice serum post immunization was collected biweekly and antigen specific neutralizing antibody levels were measured. The ELISA performed was antigen specific: we coated the 96 well plate with the antigen that was tested in each case. This assay was separately performed for all the 4 specific antigens. Elevated Antibody levels were observed for the antigens (Delta and Omicron) used in this study as well as for the alpha and beta specific antigens, where the antibody levels were low yet significant as compared to naïve. The lower levels of response in Alpha and Beta strains could be in part since the antibodies obtained from mice sera, were tested separately for the subunit part, namely the spike glycoprotein RBD of the virus obtained from Alpha and Beta strain. This is because we could only obtain the pure spike glycoprotein for testing from BEI resources. Detailed explanation is provided in the discussion part on line 594-601 of the revised manuscript.
We have mentioned in line 605 to 606, that in future studies we plan to understand the effectiveness of neutralizing antibodies in order to confirm the blocking ability of the virus to provide protection.
Specific Comments
Major points:
- Lines 164-166: Please delete these lines.
Response – We thank the reviewers for this comment we have now removed those lines.
- Line 215 & 217: Please indicate the concentration of MTT and incubation temperature so that the readers can follow your method.
Response – We have used 5mg/ml concentration of MTT reagent as stock was used and 10ul from this stock was added to each well. The incubation temperature was 370C. We have updated this in the revised manuscript on the line 225, 226.
- Line 241: Please indicate number and sex of mice used.
Response – Thank you for the comment. Six Female mice per group were used for these studies. We have included this in line 255 and 256 of revised manuscript.
- Line 260: Please make it clear about the ELISA antigens. Were they whole viruses?
Response: Thank you for your response. We have used whole inactivated virus for Omicron and Delta and Spike RBD glycoprotein for Alpha and Beta variant. Due to the lack of availability of the whole inactivated virus of Alpha and Beta strain, we had to use the spike glycoprotein part of these variants. We have included the antigens details in the line 281 of the revised manuscript.
- Lines 263-264: Please indicate the diluent for BSA.
Response: We thank reviewers for the observation. Diluent used in this experiment is Phosphate buffer saline (PBS). We have included this in manuscript on line 284.
- Figure 4: Please explain why only few bands are visible despite they are whole viruses. Can the authors include original viruses and perform western blot to confirm which band is which protein?
Response: We thank reviewers for the comment. Nucleocapsid is one of the important proteins that is present abundantly in the virus. On performing SDS PAGE analysis this band was easily detected due to the high concentration present in overall sample. In this assay the primary goal was to confirm the absence of antigen degradation during the microparticle formulation process. We wanted to demonstrate that the formulation process did not result in any antigen degradation.
Minor points:
- Line 14: “SARS-COV-2” should be “SARS-CoV-2”.
Response – Following changes are made on the line 14 of revised manuscript
- Please use proper unit for microliter “µl” but not “ul” throughput the manuscript.
Response – We have made these changes across the manuscript.
- Line 178: Please use consistent description throughput the manuscript. “mL” should be “ml”.
Response –. We have updated this in the revised manuscript.
4. Line 212: “ml” is missing after “500ug/”.
Response – We have added the term ‘ml’ on line 253 of the updated manuscript.
5. Line 231: Please delete “a plate reader using” as this is a duplication.
Response – We have removed this from the revised manuscript
6. Line 302: “In-vivo” should be “In-vitro”.
Response – Following changes are made to the revised manuscript on line 323.
7. Figure 4: Please change color of the arrows for (c) and (d).
Response - We have changed the arrow color to green and black for figure 4.
8. Line 388: “1 and 50” should be “1:50”.
Response- We have made the suggested changes on line 390 of revised manuscript.
9. Figure 9: Please change “Week” to “WEEK” and “Terminal” to “TERMINAL” for consistency.
Response – We have made the changes in the figure 9 and uploaded the revised figure in the manuscript.
Round 2
Reviewer 1 Report
Comments and Suggestions for Authors
This proof-of-concept study aimed to assess the effectiveness of microneedleassisted delivery of a heterologous microparticulate vaccine against the SARS-CoV-2 virus. This research used a heterologous vaccine strategy approach using inactivated viruses from Delta and Omicron variants, delivered via transdermal microneedle. The research team encapsulated viral antigens in PLGA microparticles with alum adjuvant. Experimental results demonstrated that microneedle microparticulate based heterologous vaccine showed cross-reactivity within the strains. However, there are some points that the authors should address that can improve the manuscript.
- The text only mentions the use of a “26-gauge needle” and a “microneedle spring applicator” but does not provide key parameters such as microneedle flow rate, contact area, and depth of penetration of injected fluids measured with a dye, which are suggested to be added.
- There are also some formatting errors,subheadings 3.3, 3.4, etc. are not indented. It is recommended to carefully review and correct these issues.
- The format of the references is not uniform, some of the references still lack DOI and the author abbreviations are inconsistent.
Author Response
Reviewer 1
1. The text only mentions the use of a “26-gauge needle” and a “microneedle spring applicator” but does not provide key parameters such as microneedle flow rate, contact area, and depth of penetration of injected fluids measured with a dye, which are suggested to be added.
Response - Thank you for the suggestion. We have now added this in the line 168-172 of revised manuscript.
‘Previous study from our lab, shows the penetration depth of 520um in porcine skin, depth of the penetration was measured by confocal microscopy using hematoxylin and eosin to visualize needle penetrability through skin layers. The dye used to measure pore forming nature was analyzed using the 1% methylene blue. Microneedles dissolve in 5min after coming in contact with the skin.’
(Braz Gomes, K.; D’Souza, B.; Vijayanand, S.; Menon, I.; D’Souza, M.J. A Dual-Delivery Platform for Vaccination Using Antigen-Loaded Nanoparticles in Dissolving Microneedles. International Journal of Pharmaceutics 2022, 613, 121393, doi:10.1016/j.ijpharm.2021.121393.)
The contact area of the microneedle is on the dorsal side of the skin surface. This is mentioned in the line 268.
2. There are also some formatting errors, subheadings 3.3, 3.4, etc. are not indented. It is recommended to carefully review and correct these issues.
Response - Thank you for the comment. We have now fixed the formatting style and heading indentation in the subheadings 3.3, 3.4 and across the manuscript.
3. The format of the references is not uniform, some of the references still lack DOI and the author abbreviations are inconsistent.
Response – Thank you for the comment. DOI is added for reference 40, 42. However, references 1,7, and 45 are official website articles where the DOI and the author's name are absent. Whereas for reference 2, DOI is absent since it doesn’t provide any information related to DOI. Reference 13 is the online book.

Reviewer 3 Report
Comments and Suggestions for Authors
The revised manuscript is much improved. The authors have responded to most of the critiques in a satisfactory way. The only critique this reviewer believes is still unanswered and affects the
scientific rigor of the work is the definition of neutralizing antibody as it is well-known that only anti-RBD antibody titers correlate neutralizing antibody titer.
Author Response
The revised manuscript is much improved. The authors have responded to most of the critiques in a satisfactory way. The only critique this reviewer believes is still unanswered and affects the
scientific rigor of the work is the definition of neutralizing antibody as it is well-known that only anti-RBD antibody titers correlate neutralizing antibody titer.
Response – Thank you for the comment.
In this study we have analyzed antibodies produced post administration of the vaccine. Our model antigen comprises of whole cell inactivated SARS-CoV-2 virus which includes the receptor binding domain and all the different epitopes of the virus, antibodies are generated against these epitopes of virus. Consequently, antibodies are produced against these viral epitopes. These antibodies perform several functions, including neutralization, opsonization, complement activation, and agglutination. Specifically, when antibodies selectively bind to the RBD, they contribute to viral neutralization. As the reviewer correctly pointed out the neutralizing nature is associated with the ability of binding towards RBD, we have now removed the word ‘neutralizing’ antibody from the manuscript. As this study focuses only on the capacity of the vaccine to produce different antibodies, in future study, we plan to analyze the functionalities of these antibodies.

Round 3
Reviewer 3 Report
Comments and Suggestions for Authors
I'm sorry but I'm still don't understand you conclusions. If you don't measure strain-specific neutralizing antibodies, how are you sure that you are inducing cross-reacting antibodies as those variants you used have around 99% amino acids homology for major proteins?
Author Response
I'm sorry but I'm still don't understand you conclusions. If you don't measure strain-specific neutralizing antibodies, how are you sure that you are inducing cross-reacting antibodies as those variants you used have around 99% amino acids homology for major proteins?
Response - We appreciate the reviewer’s critical perspective on antibody neutralization. While we acknowledge that direct neutralization assays were not performed, our whole inactivated virus vaccine approach provides significant insights into cross-variant immune responses. The near-identical (99%) amino acid homology among SARS-CoV-2 variants suggests that antibodies generated would likely show cross-reactivity. Our antigen-specific ELISA revealed antibody generation against multiple viral epitopes, demonstrating the vaccine’s potential to elicit a broad immune response. We recognize the limitations of our current study and transparently outline plans for future research to directly measure neutralizing capabilities and antibody blocking efficiency. The comprehensive characterization of variant-specific antibody responses offers valuable preliminary data that contributes to our understanding of microneedle vaccine delivery and potential cross-reactive immune mechanisms, setting the stage for more targeted neutralization studies.